# Time-Budget of Horses Reared for Meat Production: Influence of Stocking Density on Behavioural Activities and Subsequent Welfare

**DOI:** 10.3390/ani10081334

**Published:** 2020-08-01

**Authors:** Federica Raspa, Martina Tarantola, Domenico Bergero, Joana Nery, Alice Visconti, Chiara Maria Mastrazzo, Damiano Cavallini, Ermenegildo Valvassori, Emanuela Valle

**Affiliations:** 1Department of Veterinary Sciences, University of Turin, 10095 Grugliasco, Italy; federica.raspa@unito.it (F.R.); martina.tarantola@unito.it (M.T.); domenico.bergero@unito.it (D.B.); joana.nery@unito.it (J.N.); alice.visconti@edu.unito.it (A.V.); chiara.mastrazzo@edu.unito.it (C.M.M.); 2Department of Veterinary Medical Sciences, University of Bologna, 40064 Ozzano dell’Emilia, Italy; damiano.cavallini@unibo.it; 3Public Veterinary Service, ASL TO5, 10023 Chieri, Italy; e.valvassori@tin.it

**Keywords:** horse, behaviour, time-budget, welfare, stocking density

## Abstract

**Simple Summary:**

Horses reared for meat production are kept in group pens at high stocking densities. Due to the lack of scientific knowledge concerning the welfare of horses reared in this way, the aim of the present work was to assess whether their behaviours were affected by stocking density. The time-budget of the horses was also studied to evaluate if and how it differed compared with that of wild-living horses. We found that the expression of locomotion, playing, and self-grooming increased as the space allowance per horse within the group pens increased, indicating the potential to use these behaviours as indicators of positive welfare. Moreover, an altered time-budget was identified, implicating the condition of compromised welfare in these animals. Standing was the main expressed behavioural activity. A higher than usual amount of time was spent in a lying position, and a lower than usual amount of time was dedicated to feeding and locomotion. This study was the first to assess the behaviour of horses reared for meat production. The results show that more attention needs to be directed at the housing and management conditions under which horses reared for meat production are kept in in order to improve their welfare.

**Abstract:**

Horses reared for meat production can be kept in intensive breeding farms where they are housed in group pens at high stocking densities. The present study aimed to evaluate whether the expressed behaviours correlated with stocking density, and to compare their time-budget with that of wild-living horses. An ethogram of 13 mutually exclusive behavioural activities was developed. Behavioural observations were performed over a 72 h period on group pens selected on the basis of stocking density and the homogeneity of breed, age, height at the withers, and time since arriving at the farm. Scan sampling (*n* = 96 scans/horse/day) was used on 22 horses. The mean frequency (%) ± standard deviation (±SD) for each behavioural activity was calculated to obtain the time-budget. The associations between time-budget and stocking density were evaluated using a bivariate analysis. The relationships were analysed by Pearson’s correlation coefficient (*r*). Our results show that locomotion, playing, and self-grooming positively correlated with a reduction in stocking density, indicating the potential to use these behaviours as positive welfare indicators for young horses kept in group pens. The data also revealed an unusual time-budget, where the main behavioural activity expressed was standing (30.56% ± 6.56%), followed by feeding (30.55% ± 3.59%), lying (27.33% ± 2.05%), and locomotion (4.07% ± 1.06%).

## 1. Introduction

Most of the scientific literature on horses reared for meat production is focused on the final product—the meat—in terms of its consumption [1] and nutritional values [2,3]. In contrast, there is a lack of scientific studies assessing equine faming conditions and how to safeguard horse welfare. According to Faostat data [4], more than 500,000 horses are slaughtered in Europe each year. Among the European community countries, the consumption of horse meat is limited to Spain, Italy, France, and Belgium [1,2]. However, it is reported that there are no standardised farming conditions for the breeding of the horses reared for meat production [2]. What is clear is that farms breeding horses for meat production rear young horses [5], and that these animals are often kept in intensive farming systems in order to increase meat production performances [6]. Overcrowding and high stocking densities are a concern with regard to intensive livestock farming [7]. Indeed, the European Commission has recognised that increasing the space allowance for animals kept in group pens is key to improving their welfare [8].

A high stocking density can negatively affect horse welfare, threatening the horse’s physiological and behavioural needs [9]. High stocking densities lead to spatial restrictions that may prevent the animals from expressing behaviours that would otherwise be performed under more natural conditions [10]—e.g., the reduction in the expression of positive social interactions as allogrooming [11], and the reduction in the expression of feeding behaviour while exploring and moving [12,13]. The increase in the space available per animal that accompanies a reduction in a group pen stocking density has been reported to increase the expression of certain behaviours in a number of domestic species, including growing pigs [14], broiler chickens [15], and cattle [16], and is thought to reflect an improvement in their welfare state. To the best of our knowledge, no studies have evaluated to date whether an increase in the space allowance per horse kept in a group pen can generate an improvement in the behavioural indicators of positive welfare.

According to the three dimensional-concept proposed by Fraser et al. [10], which integrates the Five Freedoms [17], an animal welfare assessment needs to encompass the study of animal behaviour. This natural-living orientation represents a reference point for the Five Domains Model proposed by Mellor [18]. Accordingly, Domain 4—labelled “Behaviour”—aims at focusing attention on the environmental circumstances and their impact on the affective states experienced by animals [19]. In particular, inadequate living conditions can affect animal behaviours, leading to modifications in their time-budget and/or behavioural repertoire [20,21]. It is reported that, despite the process of domestication, horses have maintained the species-specific behaviours of their wild ancestors [22]. Studying the time-budget of horses kept in human-managed environments is, therefore, a useful tool that can help us understand their state of welfare [23].

On this basis, the first aim of the present study was to evaluate whether the behavioural activities performed by horses reared for meat production were affected by the stocking density in which they were housed. The second aim was to investigate the time-budget of horses kept in intensive breeding farms for meat production and to compare the observed time-budget with the data available in the scientific literature about wild-living horses.

## 2. Materials and Methods

The present study was approved by the Ethical Committee of the Department of Veterinary Sciences of the University of Turin, Italy (Prot. n. 2202).

### 2.1. Animals and Animal Husbandry

The present study was conducted in the biggest horse breeding farm for meat production in Northern Italy. This farm adopts intensive farming methods and sends a total of 2000 horses to slaughter each year. This farm housed around 300 young horses of 16 ± 8 months (mean ± standard deviation) for each cycle of production. The horses—of different heavy draft breeds and both sexes—were housed and managed according to typical farm conditions for meat production, and none of the conditions were altered in any way for the purposes of this research. The horses were housed in group pens in a barn with two open sides, and they had no access to any outdoor paddock area. The pens were characterised by different sizes (from 14.9 to 46.5 m^2^). On the basis of the pen size, the number of horses varied within each pen (from 2 to 15 horses) according to the choice of the breeder. Stallions and female horses were kept together. Each pen was enclosed by horizontal metal rail bars, and tap water was provided by a single automatic drinker. The floor was concrete and covered with barley straw bedding that was added daily (before the evening meal) by an automatic straw-dispersing tractor to cover the pen floor with a thickness of 15 cm of straw. More details on the housing and management conditions on this farm are provided by Raspa et al., 2020 [6].

Twice a day (at 7 am and at 6 pm) from the feeding lane the horses were supplied with long-stem first-cut meadow hay (6 kg/animal/day), plus 8 kg/animal/day of a cereal-based concentrate pelleted feed, labelled as follows (% of dry matter): crude protein 14.50%, ether extract 3.50%, crude fibre 5.70%, ash 6.60%; as fed: starch 55%.

### 2.2. Selection of Group Pens

The inclusion criteria for pen selection were based on the stocking densities. Moreover, group pens needed to be homogenous for breed, age, height at the withers, and time since arriving at the farm. This latter criterion ensured that all the horses were equally accustomed to the housing and management conditions of the breeding farm.

Stocking density was expressed as the m^2^ per horse (m^2^/horse). Once the area of each pen was recorded, it was divided by the mean height of the horses, measured to the withers, within the pen. A laser meter was used to measure the height of animals at the withers, and only pens containing same-sized animals were assessed. The space allowance at the feed bunk was calculated by dividing the length of the feed bunk (meters) by the number of horses within the pen (m/horse).

Only three group pens in the barn met these criteria. Table 1 reports the number of horses, pen area (m^2^), stocking density (m^2^/horse), and feeding space per horse at the feed bunk (m/horse) for each pen. A total amount of 22 horses (19 males and 3 females) with a height at the withers ranging between 140 and 150 cm were involved in the study. All the horses belonged to the Comtois breed, and their mean age (±standard deviation) was 22 ± 2 months. All the animals had spent six weeks in the barn before being involved in the present study.

### 2.3. Behavioural Observations

One 2D camera equipped with infrared light (Hikvision IP 3.0 Megapixel—NDV Network Video Recorder Hikvision 7600 Series) was installed on each selected pen. The cameras were oriented so that the horses were never out of sight. Observations were recorded for 72 h, corresponding to three consecutive days (24th to 26th November).

The videos were evaluated by two trained observers—experts in the equine field—using an ethogram recording sheet (Table 2). The ethogram was developed to assess 13 mutually exclusive behavioural activities, meaning that the horse could only be doing one of the named activities at any one time (as suggested by McFarland and Sibly, 1975 [24]). Before the behavioural data were collected, the observers underwent specific training to be ensure an adequate degree of concordance in how they interpreted the data. Thus, the inter- and intra-observer reliability were evaluated as indicated in the data and statistical analysis section. The observations of behavioural activities were performed using scan sampling [25,26]. The behaviours expressed by each horse in the pens were assessed by scan sampling at 15 min intervals throughout the 72 h observation period.

### 2.4. Data and Statistical Analysis

Statistical analyses were performed using JMP v14.3 (SAS Institute Inc., Cary, NC, USA). The inter- and intra-observer reliability of the trained observers was evaluated by means of the Cohen’s Kappa Coefficient (K).

Each pen was considered as a statistical unit. In order to investigate the time-budget pattern, we used the frequency (%) ± SD for the selected behavioural activities. Frequencies were calculated for each day of observation, and data were collected for:-24 h periods (%/24 h);-12 daylight hours (8:00 am–8:00 pm) (%/daylight hours);-12 night hours (8:00 pm–8:00 am) (%/night hours).

#### 2.4.1. Correlations between Time-Budget and Stocking Densities within Group Pens

Bivariate analysis was used to investigate the effect of stocking density (categorical predictors, 4, 5 and 6 m^2^/horse) on the behavioural activity frequencies (%/24 h; %/daylight hours; %/night hours). Relationships were analysed using the Pearson’s correlation coefficient (*r*, 1 or −1 depending on whether the variables are positively or negatively related [27]). The r coefficient values for correlation were interpreted according to Prior and Haerling [28]: very strong correlation (±0.91 to ±1.00); strong correlation (±0.68 to ±0.90); moderate correlation (±0.36 to ±0.67); weak correlation (±0.21 to ±0.35); and negligible correlation (0 to ±0.20). The probability of correlation (*p*-value) was calculated and Pearson correlations were considered significant at *p* ≤ 0.05.

#### 2.4.2. Overall Time-Budget and Time Frame

We calculated the mean frequency value for each behavioural activity for the 72 h observation period (overall time-budget) considering all 22 horses. The overall time-budget of each behavioural activity engaged in by the horses was further divided according to 6 time intervals (00:00–04:00; 04:00–08:00; 08:00–12:00; 12:00–16:00; 16:00–20:00; 20:00–24:00) as described by Boyd et al. [29]. In particular, data for the time-budget of the main expressed behavioural activities (feeding, lying, standing, and locomotion) performed by young Przewalski horses (age range: 2 to 3 years) were adapted from Boyd et al. [29] in order to compare the behavioural activities between horses reared for meat production and wild-living horses.

## 3. Results

The inter-observer reliability was exceptionally high: K = 0.83 (95% CI [0.72–0.94]) The intra-observer reliability was substantial K = 0.67 (95% CI [0.59–0.75]) for the first evaluator, and very high for the second evaluator K = 0.81 (95% CI [0.75–0.87]) [30].

A total amount of 96 scans per horse were performed each day, providing a total of 6336 scans sampled over the 72 h video-recordings.

### 3.1. Correlations between Time-Budget and Stocking Densities within Group Pens

The reduction in the stocking density and the subsequent increase in the space allowance per horse (from 4 to 6 m^2^/horse) was positively correlated with locomotion (*r* = 0.89, *p* = 0.001), playing (*r* = 0.73, *p* = 0.024), and self-grooming (*r* = 0.76, *p* = 0.018) (Table 3). The data obtained revealed that the reduction in stocking density correlated with a higher frequency in the expression of these activities by horses. Locomotion showed a positive correlation with the reduction in stocking density during both the 12 daylight hours (%/12 light hours) (*r* = 0.76, *p* = 0.017) and 12 night hours (%/12 night hours) (*r* = 0.67, *p* = 0.049). Playing seemed to be positively and significantly correlated with the reduction in stocking density during the 12 daylight hours (*r* = 0.79, *p* = 0.012), but not during the 12 night hours (*r* = 0.29, *p* = 0.444); the same was true for self-grooming, which showed a positive correlation during the 12 daylight hours (*r* = 0.78, *p* = 0.014), but not during the 12 night hours (*r* = 0.48, *p* = 0.193).

Although standing was not significantly correlated with stocking density over the whole 24 h period, a negative correlation was shown during the 12 night hours (*r* = −0.68, *p* = 0.049). Based on this data, the reduction in the stocking density was associated with a reduction in the expression of standing behaviour during the 12 night hours of the 24 h period.

### 3.2. Overall Time-Budget and Time Frame

As represented in Figure 1, the overall time-budget of each behavioural activity engaged in by horses reared for meat production showed that the main expressed activities were: standing (30.56% ± 6.56%), feeding (30.55% ± 3.59%), and lying (27.33% ± 2.05%). Locomotion occupied only 4.07% ± 1.06% of the time. All the other activities occupied less than the 2% of the overall time-budget. In particular, stereotypic behaviours were performed the least, occupying just 0.04% ± 0.12% of the time.

The overall time-budget of each behavioural activity shown by horses was divided into six time intervals (00:00–04:00; 04:00–08:00; 08:00–12:00; 12:00–16:00; 16:00–20:00; 20:00–24:00). As reported in Table 4, the main activity from 00:00–04:00 was lying (46.61% ± 1.19%), followed by standing (26.33% ± 4.05%), feeding (20.14% ± 2.12%), and locomotion (3.07% ± 1.63%). The time interval 04:00–08:00 showed a similar pattern, with lying being the main behaviour (51.48% ± 6.79%), followed by standing (26.01 ± 4.31%), feeding (13.43% ± 4.96 %), and locomotion (3.01% ± 0.75%). Considering the 08:00–12:00 time interval, the main activity was feeding (43.11% ± 3.65%), followed by standing (29.40% ± 6.99%), lying (10.30% ± 5.10%), and locomotion (7.38% ± 4.66%). The main activity expressed during the 12:00–16:00 time interval was standing (32.67% ± 6.93%), then feeding (31.94% ± 3.40%), lying (21.38% ± 0.93%), and locomotion (2.95% ± 0.15%). The same pattern of expression was also shown for 16:00 to 20:00, where the main expressed activity was standing (41.06% ± 1.48%), followed by feeding (38.74% ± 5.64%), locomotion (5.70% ± 4.26%), and lying (4.46% ± 2.13%). From 20:00 to 24:00, feeding was the main activity (35.94% ± 4.19%), followed by lying (29.77% ± 2.61%), standing (27.86% ± 6.64%), and locomotion (2.34% ± 1.71%).

Stereotypic behaviour was only present during the time intervals 12:00 to 16:00 and 20:00 to 24:00, although horses were only engaged in this activity for 0.12% ± 0.20% of the time.

Figure 2 shows the comparison of the 24 h time frame of the main expressed behavioural activities (standing, feeding, lying, and locomotion) performed by the horses reared for meat production and young Przewalski horses (data adapted from Boyd et al., 1988 [29]).

## 4. Discussion

Studying the behaviours of animals reared in human-managed environments and comparing their time-budgets with those of animals living in natural environments is important for understanding animal welfare in the former [20]. Despite the process of domestication, horses have maintained the species-specific behaviours of their wild ancestors [23]. Consequently, the reduction in the horse’s behavioural repertoire and/or the change in time-budget can reflect a low or inadequate welfare status [21,31].

In the present study, the daily time-budget performed by horses reared for meat production was mainly expressed by standing (30.56% ± 6.56%), feeding (30.55% ± 3.59%), and lying (27.33% ± 2.05%). Locomotion was engaged in 4.07% ± 1.06% of the time. By comparing these results with the data available in the literature about young (2–3 years old) wild-living horses, some important differences were observed. Przewalski horses spend 46.4% of the day feeding, 33.87% of the day standing, 7.4% of the day in locomotion, and 5.3% of the day lying down [29]. Duncan, in 1980 [32], reported similar data in young Camargue horses, which spend at least 56.37% of the daily time-budget engaged in feeding behaviour, 19.41% in standing behaviour, 6.97% lying down, and 5.55% of their time in locomotion, with variations according to the seasons. Taking these two studies into account, we can say that young wild-living horses have an overall time-budget in which feeding is the main expressed behavioural activity, followed by standing, lying, and locomotion. On the contrary, the daily time-budget of the horses of the present study reared for meat production involved standing as the main expressed behavioural activity, followed by feeding, lying, and locomotion. It seems that the environmental constraints imposed by the breeding farm resulted in these horses lying down more and moving less compared with Przewalski and Camargue horses.

The strong reduction in the expression of feeding behaviour is in accordance with the studies conducted by Yarnell et al., 2015 [33], and Benhajali et al., 2008 [31]. In the study by Yarnell et al., 2015 [33], horses housed in groups in a paddock area poor in grass spent 34.89% ± 14.3% of the time expressing feeding behaviour. As suggested by the same authors, this result was the consequence of the limited availability of grass. Moreover, in the study by Benhajali et al., 2008 [31], mares densely housed in paddocks were found to engage in feeding behaviour for 25.83% ± 26.80% of their time. These authors correlated this result with the lack of foraging opportunity. According to these two studies, our results could be interpreted in the same way, since animals were fed just twice a day with approximately 6 kg of hay/animal/day.

The reduction in feeding behaviour could also be linked to the lack of adequate space at the feed bunk, as shown in studies on other livestock species [34]. To this regard, the Code of Practice for the Care and Handling of Equines [35] recommends guaranteeing at least 1 m feeding space per horse under group-housing conditions and suggests having an extra feeding point available (i.e., one feeding point more than the number of horses). As shown in Table 1, none of the pens involved in the present study respected this indication.

The time spent standing by horses reared for meat production—30.56% ± 6.56%—was comparable with those reported in Przewalski horses at 33.87% [29]. In particular, our results show that a reduction in stocking density correlates with a reduction in the expression of standing behaviour during the night hours (*r* = −0.68, *p* = 0.049).

The time-budget of our study relating to lying behaviour—27.33% ± 2.05%—is in stark contrast with the data shown for wild-living horses. Yarnell et al., 2015 [33], reported their horses to spend just 0.08% ± 0.1% of the time lying down; and the mares studied by Benhajali et al., 2008 [31], never exhibited lying behaviour. From our results, it seems that the smaller pen areas may encourage horses to lie down more, also because locomotion behaviour was found to increase as space availability increased. The reduction in the expression and/or the absence of lying behaviour is widely recognised as a sign of reduced welfare in domestic species [36,37]. However, little is known about the normal lying behaviours of horses over the course of 24 h periods, or about what factors affect lying in horses [38]. Heleski et al. [39] suggested that an increase in lying behaviour in weanlings housed in stalls could be due to boredom and the lack of possibility to perform other behaviours. Boredom and physical restriction may also be the reason for the high frequency of lying behaviour in the horses of our study. Moreover, in the present study no correlation was found between stocking density and lying behaviour frequency. Indeed, the overall increase in space allowance per horse was probably too small to allow for any differences. In fact, no guidelines or regulations are presently available for the housing and management conditions of horses reared for meat production. The only official document issued by the EU in relation to horse welfare is the Animal Welfare Indicators (AWIN) assessment protocol for horses [40]. This document is not specific for this category of horse, but it does provide indications about the space allowance for horses kept in group housing systems. In particular, horses with a height at the withers ranging from 140 to 150 cm—as those involved in our study—require at least 7 m^2^/horse. None of the pens respected this indication. As a consequence, the limitation of this present study was related to the fact that it was not possible to have a control group in which the minimum space requirement considered by AWIN was satisfied. Moreover, only one camera per pen was used, even if the camera were oriented in order that horses were never out of sight. Interestingly, the reduction in the stocking density within the group pens positively correlated with an increase in behavioural activities such as locomotion, playing, and self-grooming. Thus, having more space available allowed the horses to move and play more; these results are in accordance with studies carried out on other domestic species (e.g., dairy calves [41] and growing pigs [42]).

Increased active locomotion (e.g., active walk, trot, and canter) has been identified in relation to inappropriate housing conditions [31,43]. However, in our study, the increase in space per animal was correlated with an increase in the expression of slow walking and explorative behaviour (sniffing the ground whilst walking; see Table 2).

Playing behaviour and self-grooming have been identified as potential positive welfare indicators in many species [44,45,46]. In particular, although growing evidence suggests that an increase in playing behaviour in adult domestic horses could be related to inappropriate living conditions [47], it seems that young horses only express playing behaviour under favourable breeding conditions [21]. Therefore, an increase in playing behaviour according to an increase in the space available could be considered as a positive welfare indicator in young horses.

Since grooming is reported to be an expression of horse welfare [48], the increase in self-grooming according to the increase in the group pen space allowance may be linked to improved welfare and could be proposed as a positive welfare indicator in this kind of breeding farm. However, the significance of self-grooming as a positive behaviour is less clear than that of mutual grooming. In fact, it seems that when horses are kept in a group, they engage more in mutual grooming [44]. However, it has also been suggested that the performance of self-grooming could be a sign of increased welfare (being a rewarding behaviour), as proposed for mutual grooming [44].

All the other behavioural activities occupied less than 7.49% of the total daily time-budget. The particularly low frequency of stereotypic behaviour is interesting to note. It is well known that an increased frequency of stereotypic behaviour may correspond with an animal’s attempt to cope with an inadequate environment [49]. However, as a result of the imposed management conditions—i.e., the high stocking densities, the feeding regime used, and the impossibility to perform free movement—standing was the main expressed daily behavioural activity. Fureix et al., 2012 [50], showed that horses living under unfavourable welfare conditions can show apathy and unresponsiveness to environmental stimuli. Although in the present work it was not possible to study body position, in order to identify the apathetic state, the poor expression of stereotypic behaviours may be linked to a depressive state in these animals. The occurrence of stereotypic behaviours represents one of the most recognised behavioural indicators of welfare impairments. It could be supposed that the unusually low presence of stereotypic behaviours in horses reared for meat production could similarly reflect a condition of poor welfare. Further investigations are needed to elucidate the significance of this unexpectedly low incidence of stereotypic behaviours. Moreover, future research should investigate the importance of safeguarding the welfare of horses reared for meat production. This can also lead to differences in meat quality traits, as reported in other livestock species [51,52], but above all it would improve the quality of life of these animals.

## 5. Conclusions

Considering the different factors that could affect the time-budget of horses, the reduction in stocking density had a positive impact on the expression of some behaviours, such as locomotion, playing, and self-grooming, which could be proposed as indicators of positive welfare in young horses kept in group pens. Differences in the time-budget of horses reared for meat production were found by comparing the data with those from studies on young wild-living horses (in which the main behavioural activity performed is standing). The horses reared for meat production expressed an unusual time-budget, since, compared with wild-living horses, significantly more time was spent lying down and less time was dedicated to feeding and locomotion activities. This present study stimulates further scientific studies to improve the welfare of horses reared for meat production and to obtain insight into relationships between animal welfare and meat quality, since this latter aspect represents a powerful tool to generate changes in horse meat industry practices.

## Figures and Tables

**Figure 1 animals-10-01334-f001:**
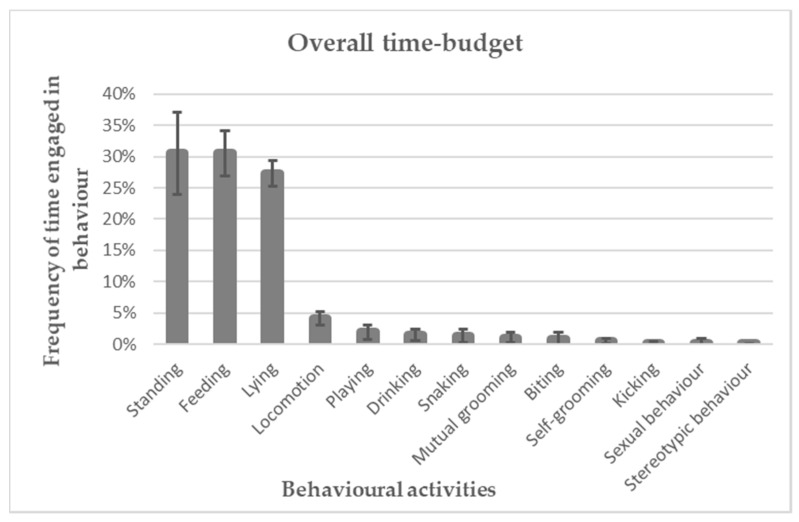
Frequency of time (%) spent by horses in behavioural activities.

**Figure 2 animals-10-01334-f002:**
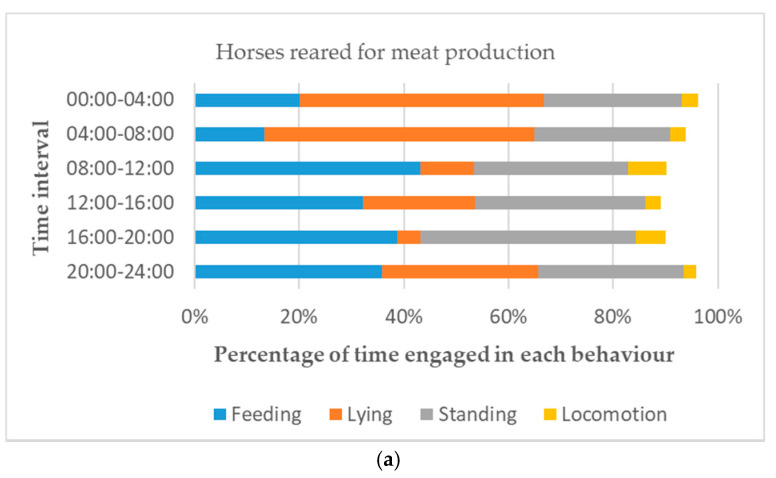
Comparison of the 24 h time frame of the main expressed behavioural activities (standing, feeding, lying, and locomotion) engaged in by the horses reared for meat production (**a**) and wild-living Przewalski horses (**b**) (data adapted from Boyd et al., 1988 [29]).

**Table 1 animals-10-01334-t001:** The number (*N*) of horses, pen area (m^2^), stocking density (m^2^/horse), and space at the feed bunk (m/horse) within each pen are reported.

Id Pen	*N* of Horses	Pen Area (m^2^)	Stocking Density (m^2^/horse)	Space at the Feed Bunk (m/horse)
A	8	35.00	4	0.88
B	8	36.75	5	0.61
C	6	36.00	6	0.80

**Table 2 animals-10-01334-t002:** Description and illustrations of the selected mutually exclusive behaviour activities.

Activities	Descriptions	Illustrations
Self-grooming	The horse performs body cleaning by himself. It includes: shaking the entire body or a part of it (a); nibbling or licking the coat hair (b); rolling on the ground (c); rubbing parts of the body against objects (d) or other parts of the body (e.g., rubbing the muzzle against the limbs) (e).	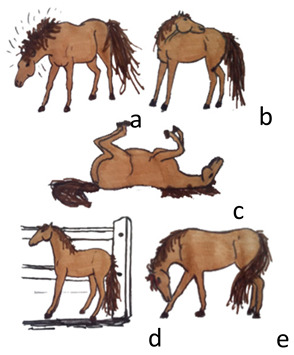
Mutual grooming	Body cleaning is performed reciprocally or by one horse towards a conspecific.	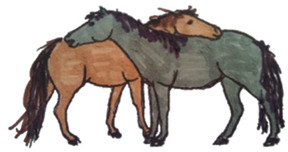
Lying	The horse is lying on the ground in the sternal position with the limbs flexed below the body (f) or in lateral position with extended limbs (g).	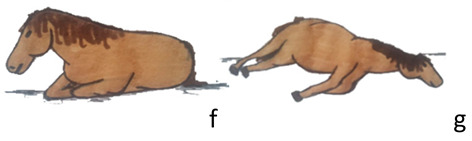
Playing	The horse plays alone or with other horses. It includes: play with structural parts of the pen (h), sexual play (i), locomotor play (l), and play fighting (m).	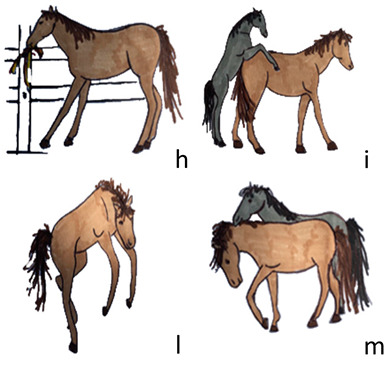
Locomotion	The horse moves inside the pen by taking steps; the neck is in a horizontal position (n) or lowered to the ground to sniff (o).	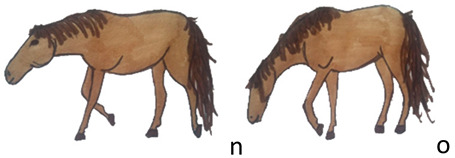
Feeding	The horse eats hay, straw or feedstuff in the trough or on the ground.	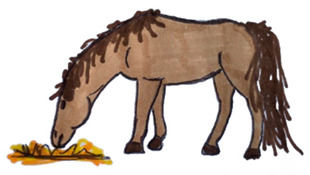
Drinking	The horse drinks.	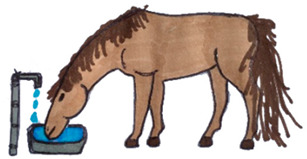
Standing	The horse is in quadrupedal station. The expression is relaxed or attentive. It includes: “standing alert” (p) and “standing relaxed” (q).	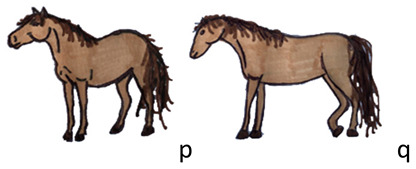
Snaking	The horse stretches its neck towards a conspecific with the ears turned backwards, the lips are often closed and the body is in a dominant position.	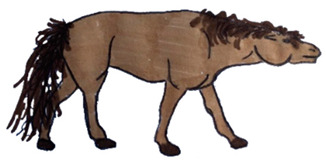
Kicking	The horse lifts one (r) or both hind limbs (s) off the ground and quickly stretches it/them towards a conspecific, aiming to hit him.	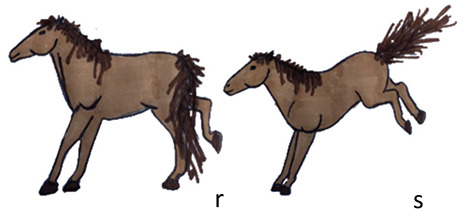
Biting	The horse quickly opens and closes its mouth and its teeth touch the body of a conspecific, aiming to bite him. The ears are turned backwards.	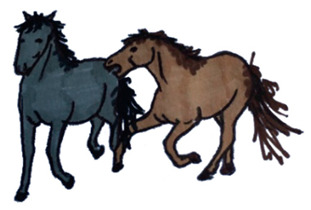
Sexual behaviour	The stallion sniffs or bites the mare’s genitals (t). The stallion mounts the mare: erection and penetration are present (u).	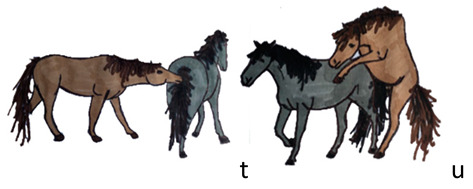
Stereotypic behaviour	The horse expresses a stereotyped behaviour: both oral (v) and locomotor stereotypes (z) are considered.	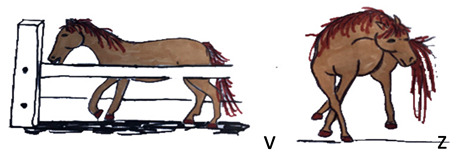

**Table 3 animals-10-01334-t003:** Associations between the time-budgets (%/24 h; %/12 light hours; %/12 night hours) and stocking densities among the group pens.

Behavioural Activities	Stocking Density					
	%/24 h		%/12 Light Hours		%/12 Dark Hours	
	*r* ^a^	*p*-Value	*r* ^a^	*p*-Value	*r* ^a^	*p*-Value
Standing	−0.61	0.079	−0.51	0.157	−0.68	0.049 *
Feeding	−0.23	0.559	−0.14	0.724	−0.32	0.396
Lying	0.59	0.094	−0.08	0.839	0.57	0.112
Locomotion	0.89	0.001 *	0.76	0.017 *	0.67	0.049 *
Playing	0.73	0.024 *	0.79	0.012 *	0.29	0.444
Drinking	−0.29	0.450	−0.56	0.114	0.00	0.997
Snaking	0.23	0.553	0.28	0.461	0.04	0.911
Mutual grooming	0.29	0.449	0.28	0.473	0.17	0.659
Biting	0.36	0.346	0.29	0.450	0.39	0.301
Self-grooming	0.76	0.018 *	0.78	0.014 *	0.48	0.193
Kicking	0.35	0.361	0.37	0.330	0.10	0.807
Sexual behaviour	0.38	0.317	0.39	0.297	0.00	1.000
Stereotypic behaviour	0.43	0.244	0.43	0.244	0.43	0.244

^a^ Pearson’s correlation coefficient. * Statistical significance *p* < 0.05

**Table 4 animals-10-01334-t004:** Overall time-budget and time frames of different behavioural activities performed by horses reared for meat production. Frequencies (%) of behavioural activities are expressed as means ± SD.

Behavioural Activities (%)	Overall Time-Budget	00:00–04:00	04:00–08:00	08:00–12:00	12:00–16:00	16:00–20:00	20:00–24:00
Standing	30.56 ± 6.56	26.33 ± 4.05	26.01 ± 4.31	29.40 ± 6.99	32.67 ± 6.93	41.06 ± 1.48	27.86 ± 6.64
Feeding	30.55 ± 3.59	20.14 ± 2.12	13.43 ± 4.96	43.11 ± 3.65	31.94 ± 3.40	38.74 ± 5.64	35.94 ± 4.19
Lying	27.33 ± 2.05	46.61 ± 1.19	51.48 ± 6.79	10.30 ± 5.10	21.38 ± 0.93	4.46 ± 2.13	29.77 ± 2.61
Locomotion	4.07 ± 1.06	3.07 ± 1.63	3.01 ± 0.75	7.38 ± 4.66	2.95 ± 0.15	5.70 ± 4.26	2.34 ± 1.71
Playing	1.97 ± 1.16	0.58 ± 1.00	1.56 ± 0.90	3.36 ± 3.03	3.13 ± 1.04.	3.04 ± 2.12	0.17 ± 0.30
Drinking	1.51 ± 0.86	1.22 ± 0.91	1.19 ± 0.58	1.59 ± 1.18	2.03 ± 1.10	0.90 ± 0.62	2.17 ± 0.66
Snaking	1.27 ± 1.07	0.43 ± 0.54	1.33 ± 0.99	2.08 ± 0.90	1.24 ± 1.04	2.11 ± 1.72	0.43 ± 0.54
Mutual grooming	1.07 ± 0.85	0.69 ± 1.20	1.04 ± 0.90	1.01 ± 0.95	1.74 ± 1.59	1.56 ± 1.38	0.38 ± 0.39
Biting	0.84 ± 1.00	0.43 ± 0.54	0.78 ± 1.14	0.81 ± 0.56	1.22 ± 1.08	1.50 ± 1.12	0.29 ± 0.27
Self-grooming	0.52 ± 0.37	0.49 ± 0.43	0.17 ± 0.30	0.64 ± 0.70	0.93 ± 0.70	0.49 ± 0.22	0.41 ± 0.36
Kicking	0.19 ± 0.22	0.00 ± 0.00	0.00 ± 0.00	0.20 ± 0.18	0.55 ± 0.74	0.26 ± 0.26	0.12 ± 0.20
Sexual behaviour	0.07 ± 0.08	0.00 ± 0.00	0.00 ± 0.00	0.12 ± 0.20	0.12 ± 0.20	0.17 ± 0.15	0.00 ± 0.00
Stereotypic behaviour	0.04 ± 0.12	0.00 ± 0.00	0.00 ± 0.00	0.00 ± 0.00	0.12 ± 0.20	0.00 ± 0.00	0.12 ± 0.20

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
