# Peer review of "Time-Budget of Horses Reared for Meat Production: Influence of Stocking Density on Behavioural Activities and Subsequent Welfare"

_animals, 2020, doi:10.3390/ani10081334_

Round 1
Reviewer 1 Report
An interesting study. There is a need for greater clarity in how you describe some of the results – as in parts it is a little confusing for the reader.
General
Specific comments
Line 37 – what is c in brackets – usual convention for correlation is R and R2 for regression Also it is pearsons product moment and spearmans rank correlation
Line 47 – replace by contrary with … In contrast, …..
Line 89 – consider restructuring the sentence … Twice a day (at 7am and 6 pm) from the feeding lane the horses were supplied with long stem first cut meadow hay…….
Lines 137-141 did you use pearsons correlation or a rank correlation test?
Lines 157-160 your values and statements do not align – if positively correlated then as stocking density increases then so does the frequency of locomotor behaviour – which is what the positive values indicates – please rewrite and clarify
Author Response
Dear Reviewer,
We have answered every concern point by point. The answers are visible in red.
We attach the file where the changes are visible in red.
Thank you so much for your valuable comments and your effort to improve the quality of our manuscript!
An interesting study. There is a need for greater clarity in how you describe some of the results – as in parts it is a little confusing for the reader.
General
Thank you for your comment. We agree with your point. We have answered to your specific comments on clarity of the results in the answer below. (Lines 174-180)
Specific comments
Line 37 – what is c in brackets – usual convention for correlation is R and R2 for regression Also it is pearsons product moment and spearmans rank correlation
We have provided to change C in r as suggested by reviewer 2, since we intended the pearson’s correlation coefficient. (Line 37)
Line 47 – replace by contrary with … In contrast, …..
Replaced as you suggested. (Line 47)
Line 89 – consider restructuring the sentence … Twice a day (at 7am and 6 pm) from the feeding lane the horses were supplied with long stem first cut meadow hay…….
Thank you. Sentence is now reworded according to your suggestion. (Line 102)
Lines 137-141 did you use pearsons correlation or a rank correlation test?
We have used pearson’s correlation. Now sentence is rewritten according also to the suggestions provided by reviewer 2 (Lines 153-158)
Lines 157-160 your values and statements do not align – if positively correlated then as stocking density increases then so does the frequency of locomotor behaviour – which is what the positive values indicates – please rewrite and clarify
Thank you for your valuable comment. Probably sentence was not clear, now it is rewritten. (Lines 174-180)

Reviewer 2 Report
Thank you for undertaking and submitting this work; horses so often get overlooked as a production species and it is essential that research such as yours is conducted to ensure we manage and protect the welfare of horses within the meat industry. This is a really interesting study which will be of great interest to readers, with some minor amendments.
Simple summary: clear and summarises study well, suitable for audience
Abstract: very good summary of study provided which is clearly differentiated from simple summary; there is scope to consider implications of results more but appreciate word count is limiting
Key words: stocking is incorrectly spelt
Introduction:
Really enjoyed reading your introduction, establishes strong rationale for study and sets up the reader less familiar with the topic well. I did wonder if it would be worthwhile including brief details of the equine meat industry e.g. which countries etc but this would be at the authors’ discretion.
Line 56: may be beneficial to provide a couple of examples of which behaviours are / may be observed at reduced frequencies in these conditions
Line 64: it would be worthwhile to also consider Mellor’s 5 domains model in here – links well to the premise of your study
Materials and methods:
Well described, repeatable methods presented.
Line 80: please amend send to sends
Line 81: for readers less familiar with equine meat production systems it would be beneficial to outline what typical farm conditions are – these are outlined in next paragraph but an average number of horses, size of pen, depth of bedding etc would give readers more context of the horses living conditions
Line 107: please remove moreover, not needed here
Table 2: nice inclusion
Line 127: I would suggest just calling this section data analysis as manipulation can have adverse connotations
Line 137: it would be useful to define your interpretation of r coefficient values for correlations in the methods aligned to a relevant citation
Results:
Generally well presented and clearly interpreted for the reader.
Line 156: I would suggest changing C (correlation co-efficient) to r throughout this section as r is more widely recognised as the correlation coefficient abbreviation
Discussion:
Relevant discussion of key results presented, with reference to appropriate research. One area which the authors may like to consider including in broader discussion and linking to future research is the potential impact of their results on meat quality as many studies in other species have reported relationships between this and animal welfare – and this can be a powerful tool to generate change in industry practices.
Please include discussion of limitations within your work and potential impact e.g. only having one camera per pen. Could also consider next steps for this research field.
Line 263: please remove date for Heleski citation
Conclusions:
would be nice to also consider or highlight some future research directions here
Author Response
Dear Reviewer,
We have answered every concern point by point. The answers are visible in red.
We attach the file where the changes are visible in red.
Thank you so much for your valuable comments and your effort to improve the quality of our manuscript!
Thank you for undertaking and submitting this work; horses so often get overlooked as a production species and it is essential that research such as yours is conducted to ensure we manage and protect the welfare of horses within the meat industry. This is a really interesting study which will be of great interest to readers, with some minor amendments.
Thank you so much to appreciate our work. Your opinion is really important for our research studies. Thank you also for your efforts to help us to improve our work.
Simple summary: clear and summarises study well, suitable for audience
Thank you!
Abstract: very good summary of study provided which is clearly differentiated from simple summary; there is scope to consider implications of results more but appreciate word count is limiting
Thank you for your comment.
Key words: stocking is incorrectly spelt
Corrected. (Line 43)
Introduction:
Really enjoyed reading your introduction, establishes strong rationale for study and sets up the reader less familiar with the topic well. I did wonder if it would be worthwhile including brief details of the equine meat industry e.g. which countries etc but this would be at the authors’ discretion.
Thank you for your comment. We have provided to add some information more about horse meat industry. (Lines 48-52)
Line 56: may be beneficial to provide a couple of examples of which behaviours are / may be observed at reduced frequencies in these conditions
We have provided to add examples according to your suggestion. (Lines 61-63)
Line 64: it would be worthwhile to also consider Mellor’s 5 domains model in here – links well to the premise of your study
Thank you for your suggestion. Sentence about 5 Domains Model is now added in Introduction. (Lines 71-73)
Materials and methods:
Well described, repeatable methods presented.
Thank you to appreciate our work!!!
Line 80: please amend send to sends
Corrected. (Line 89)
Line 81: for readers less familiar with equine meat production systems it would be beneficial to outline what typical farm conditions are – these are outlined in next paragraph but an average number of horses, size of pen, depth of bedding etc would give readers more context of the horses living conditions
Thank you for this suggestion. More information is now provided about farm conditions. (Lines 90-101)
Line 107: please remove moreover, not needed here
Done (Line 120)
Table 2: nice inclusion
Thank you so much!!!
Line 127: I would suggest just calling this section data analysis as manipulation can have adverse connotations
Ok. We have delated manipulation according to your comment. (Line 140)
Line 137: it would be useful to define your interpretation of r coefficient values for correlations in the methods aligned to a relevant citation
Ok. We have provided to add interpretation of r coefficient values for correlations in method section. (Lines 153-158)
Results:
Generally well presented and clearly interpreted for the reader.
Thank you for your comment.
Line 156: I would suggest changing C (correlation co-efficient) to r throughout this section as r is more widely recognised as the correlation coefficient abbreviation
We agree with you. We had provided to change C to r. (Lines 174-187 and Table 3)
Discussion:
Relevant discussion of key results presented, with reference to appropriate research. One area which the authors may like to consider including in broader discussion and linking to future research is the potential impact of their results on meat quality as many studies in other species have reported relationships between this and animal welfare – and this can be a powerful tool to generate change in industry practices.
Please include discussion of limitations within your work and potential impact e.g. only having one camera per pen. Could also consider next steps for this research field.
Thank you for the suggestion. We are total agree about the link between animal welfare and meat quality can be a powerful tool to generate change in industry practices. We have included that important suggestion in discussion and conclusion (Lines 330-332 and Lines 341-344)
Limitations of the study are now added (Lines 292-295)
Line 263: please remove date for Heleski citation
Done. (Line 281)
Conclusions:
would be nice to also consider or highlight some future research directions
Thank you for this suggestion. Sentence about future research is now added in the conclusion section. (Lines 341-344)

Round 2
Reviewer 1 Report
The edits have improved the manuscript.
Reviewer 2 Report
Thank you for submitting your revised manuscript and highlighting your changes, this has made reviewing very easy. The additional material included has enhanced your work and I am confident your paper will make a valuable contribution to this field.
One minor tweak to address in line 100: please amend detailed to details